# Direct in-situ imaging of electrochemical corrosion of Pd-Pt core-shell electrocatalysts

Fenglei Shi[1,7], Peter Tieu ⬡[2,7], Hao Hu[1,7], Jiaheng Peng[1], Wencong Zhang[1], Fan Li[1], Peng Tao ⬡[1], Chengyi Song ⬡[1], Wen Shang ⬡[1], Tao Deng ⬡[1], Wenpei Gao ⬡[1,3] ✉, Xiaoqing Pan ⬡[4,5] ✉ & Jianbo Wu ⬡[1,3,6] ✉

Corrosion of electrocatalysts during electrochemical operations, such as low potential · high potential cyclic swapping, can cause significant performance degradation. However, the electrochemical corrosion dynamics, including structural changes, especially site and composition specific ones, and their correlation with electrochemical processes are hidden due to the insufficient spatial-temporal resolution characterization methods. Using electrochemical liquid cell transmission electron microscopy, we visualize the electrochemical corrosion of Pd@Pt core-shell octahedral nanoparticles towards a Pt nano-frame. The potential-dependent surface reconstruction during multiple continuous in-situ cyclic voltammetry with clear redox peaks is captured, revealing an etching and deposition process of Pd that results in internal Pd atoms being relocated to external surface, followed by subsequent preferential corrosion of Pt (111) terraces rather than the edges or corners, simultaneously capturing the structure evolution also allows to attribute the site-specific Pt and Pd atomic dynamics to individual oxidation and reduction events. This work provides profound insights into the surface reconstruction of nanoparticles during complex electrochemical processes.

Platinum (Pt)-based catalysts have played an irreplaceable role in electrocatalysis, especially related to energy conversions such as fuel cell[1], water splitting[2], and carbon dioxide conversion[3]. During an electrochemical redox process, bond breaking and formation in small molecules occur on the surface of electrocatalysts at the redox potentials, the microenvironments also change the electrocatalyst surface. Although the core-shell architecture could significantly reduce the Pt usage, improve durability, and offer to tune the electronic structure toward better specific electrochemical activities[4], the chemical and structural changes in the surface cause performance degradation[5,6]. The degradation is exacerbated on the highly active catalysts, including shaped and bimetallic Pt-based nanocrystal

M@Pt (M=Pd, Au, Ag, Cu, Ni, etc.) core-shell electrocatalysts[4,7–10]. Understanding the corrosion behaviors in the microenvironment during electrocatalysis is the first step toward designing more stable catalysts. Among the various ex-situ methods to study the structural evolution of Pt-based electrocatalysts during operation (e.g., cyclic voltammetry)[6,11–17], transmission electron microscopy (TEM) offers spatial resolution down to the atomic scale. In combination with aberration correction and electron energy-loss spectroscopy (EELS), the compositional segregation of shaped Pt-based alloy nanoparticles under multiple cyclic potential swaps can be visualized at close to the atomic level[6]. However, spectroscopic and scattering methods usually only offer the averaged structure and property information of all

[1]Center of Hydrogen Science & State Key Laboratory of Metal Matrix Composites, School of Materials Science and Engineering, Shanghai Jiao Tong University, 800 Dongchuan Rd, Shanghai 200240, People's Republic of China. [2]Department of Chemistry, University of California, Irvine, Irvine, CA 92697, USA. [3]Future Material Innovation Center, Zhangjiang Institute for Advanced Study, Shanghai Jiao Tong University, Shanghai 200240, People's Republic of China. [4]Department of Materials Science and Engineering, University of California, Irvine, Irvine, CA 92697, USA. [5]Department of Physics and Astronomy, University of California, Irvine, Irvine, CA 92697, USA. [6]Materials Genome Initiative Center, Shanghai Jiao Tong University, Shanghai, People's Republic of China. [7]These authors contributed equally: Fenglei Shi, Peter Tieu, Hao Hu. ✉e-mail: gaowenpei@sjtu.edu.cn; xiaoqinp@uci.edu; jianbowu@sjtu.edu.cn

catalyst particles and ex-situ experiments do not capture the structural changes in the catalysts under native operation conditions, leaving observation gaps in both the spatial and temporal resolution. Revealing the oxidative and reductive evolution in real-time on a single particle during each elementary CV cycle is key to truly reflecting the corrosion behavior of nanoparticles under low-high cyclic potentials, but remains challenging. It is essential to record the evolution of catalysts under realistic operation conditions using appropriate and advanced in-situ/operando technologies[14–17] to better understand the dynamic process in electrochemical corrosion.

With the rapid development of in-situ liquid cells in TEM[18–21], reactions in liquid environments can be recorded with high spatial resolution[22–28]. Recently, we have reported the nanoscale corrosion of different Pd@Pt core-shell electrocatalysts[5,29] under continuous oxidative etching of electrocatalysts under high potential. However, the operation and test conditions, including long-term potential-swapping like accelerated durability test (ADT), are more complicated and could lead to dissolution[30], ripening[31], and segregation[6] in Pt-based electrocatalysts[32]. Whether the process causes performance decay or improvement depends on the newly formed surface structure[6,33–36]. Identifying, predicting, and evaluating the structure-properties correlation of electrocatalysts in proton exchange membrane fuel cells (PEMFCs)[37,38] requires the point-to-point correlation between structural dynamics and performance during electrocatalysis. By coupling the in-situ liquid cell TEM (LC-TEM)[39] with an electrochemical workstation[40], the real-time structure evolution of catalysts during electrocatalysis can be monitored[21,41–43]. Several such works were performed under operating environments[44–47]. Nevertheless, most of the research papers focused on structural evolutions like coarsening, selective dissolution, and re-deposition of Pt-based nanoparticles (NPs) without further interpretation and correlative performance evaluation[41,44,48].

Herein, we utilized in-situ electrochemical LC-TEM (ELC-TEM)[49,50] to study the nanoscale structural evolution of a typical Pt-based electrocatalyst, the Pd@Pt core-shell octahedral nanoparticle, a model system with high electrochemical surface area (ECSA) and high activities toward multiple electrocatalytic reactions[4,51–53]. In this work, a single complete CV with a full range from hydrogen ad/desorption to double electric layer and oxygen ad/desorption was carried out inside TEM, showing a typical CV curve with pronounced oxidative and reductive peaks. The near-surface reconstruction of the Pd@Pt electrocatalyst in individual CV cycles was captured at nano-scale. The unprecedented structure details and change in element distribution reveal that the surface reconstruction involves preferential oxidative etching of external Pd from terrace site and reductive re-deposition of the Pd and Pt atoms to the exterior terrace surface, driven by the hydroxide/oxygen adsorbates. When coupled with an ex-situ stability test, results show the sequential corrosion and redeposition of the core-shell electrocatalysts lead to the formation of PtPd nanoframe during the CV activation.

## Results

### In Situ Investigation on electrochemical corrosion of Pd@Pt octahedron nanoparticles

A schematic diagram of the electrochemical liquid cell (ELC) is shown in Fig. 1a. 0.1 M HClO$_4$ solution was used as electrolyte and was sealed between two E-chips to form a 500 nm thick electrochemical cell. The bottom E-chip incorporates a three-electrode system, consisting of a glassy carbon working electrode (WE), a Pt reference electrode (RE), and a Pt circular counter electrode (CE) (Fig. 1b). Figure 1c shows the dispersion of Pd@Pt octahedral nanoparticles of uniform sizes on the Si$_x$N$_y$ window. The nanoparticles were synthesized by optimizing our previous methods (Supplementary Fig. 1)[4]. XRD patterns (Supplementary Fig. 2) confirmed the face-centered cubic (FCC) structure of the octahedral nanoparticles. In the atomic resolution HAADF-STEM

image of a single particle (Fig. 1d), two atomic layers of Pt are observed to epitaxially grow on the Pd octahedral core. The core-shell structure and compositional distribution were further confirmed by the elemental mapping using energy dispersive spectra (EDS) in Fig. 1e–g.

In the in-situ ELC experiment, we first performed the CV using a wide potential range from −0.5 V to 0.5 V (vs. Pt) to monitor the structure evolution. The time-series TEM images in Fig. 1h show the electrochemical corrosion process of several Pd@Pt nanoparticles (the entire corrosion process was shown in Supplementary Fig. 3 and Supplementary Movie 1) in more than 200 s. From 0 s to 97 s, some clusters grew around and on the surface of the nanoparticles. Finally, the nanoparticles lost their original morphology and eventually agglomerated (199 s). During this process, because we applied much higher potential than the redox potential of Pt$^{2+}$/Pt, both Pd and Pt atoms were first dissolved into the electrolytes in the form of Pd$^{2+}$ and Pt$^{2+}$, and then reduced and deposited around the particles and on the window together. HAADF-STEM and EDS characterizations (Supplementary Fig. 4) of the samples after this in-situ CV confirmed that the irregular nanoclusters were PdPt alloy nanoparticles. Note that the fast corrosion due to the high potential makes it hard to track the details of the entire reaction. Hence, we performed different potential ranges of CV curves to adjust and observe the reaction dynamics.

### Dynamic monitoring of the electrochemical process

Figure 2 shows the quantitative analysis of the electrochemical corrosion process when the different CV ranges are applied. A smaller CV range from −0.2 V to 0.2 V (vs. Pt) was carried out. With the lower potential applied, the slower corrosion rate allows for clear observation of the etching process. Time-series TEM images in Fig. 2a, Supplementary Fig. 5, and Supplementary Movie 2 show a slow morphological evolution during the CVs. At 195 s, multiple small islands (labeled by red and blue triangles) grew on the surface of the nanoparticles (380 s). A new island (marked by the second red triangle) was observed form at 605 s, as well as another one on the corner (labeled by green triangle). The neighboring islands (marked by two red triangles) merged and grew larger at 765 s. The formation and merging of the surface islands and the entire process was much slower than that under the wider range of potential cycling (Fig. 1h). HAADF-STEM and EDS characterizations of the nanoparticles (Supplementary Fig. 6) after the in-situ electrochemical corrosion indicated that the surface islands in Fig. 2a were small Pt nanoparticles, and the internal Pd atoms were dissolved after the loss of protection from Pt layers.

Figure 2a demonstrates that the applied potential was sufficiently high for Pt islands to corrode and grow on the surface, and subsequent corrosion of the inner Pd atoms could occur. However, note that many electrochemical reactions, such as the ORR, typically operate at lower potentials than the Pt oxidation potential. In most electrocatalytic systems, Pt exhibits sufficient corrosion resistance, with only minimal corrosion occurring even over extended periods of time[14,15,54]. The loss of activity observed in many Pt-based core-shell catalysts was due to the corrosion of the core, resulting in performance degradation[33–35]. Therefore, to investigate the corrosion process of inner Pd, we adjusted the CV potential range, setting it from -0.9 V to -0.2 V (vs. Pt). Before the experiment, a short period of CV from −0.2 V to 0.2 V (vs. Pt) was applied to create some surface Pt islands. The corrosion of inner Pd took place afterward. It is worth noting that the fuel cells typically operate at much higher voltages, especially during the startup or shutdown conditions. To further study the corrosion dynamics of the Pd@Pt core-shell structure, we selected a representative single nanoparticle (marked with green dashed diamonds) to study its detailed structure evolution within a single circle of the refined CV with the applied potential starting from −0.2 V to −0.9 V and back to −0.2 V within 280 seconds (Fig. 2b, Supplementary Fig. 7 and Supplementary Movie 3). Following the pre-CV process, Island $a$ formed on the particle surface (red dot). The growth trajectory of Island $a$ under the CV

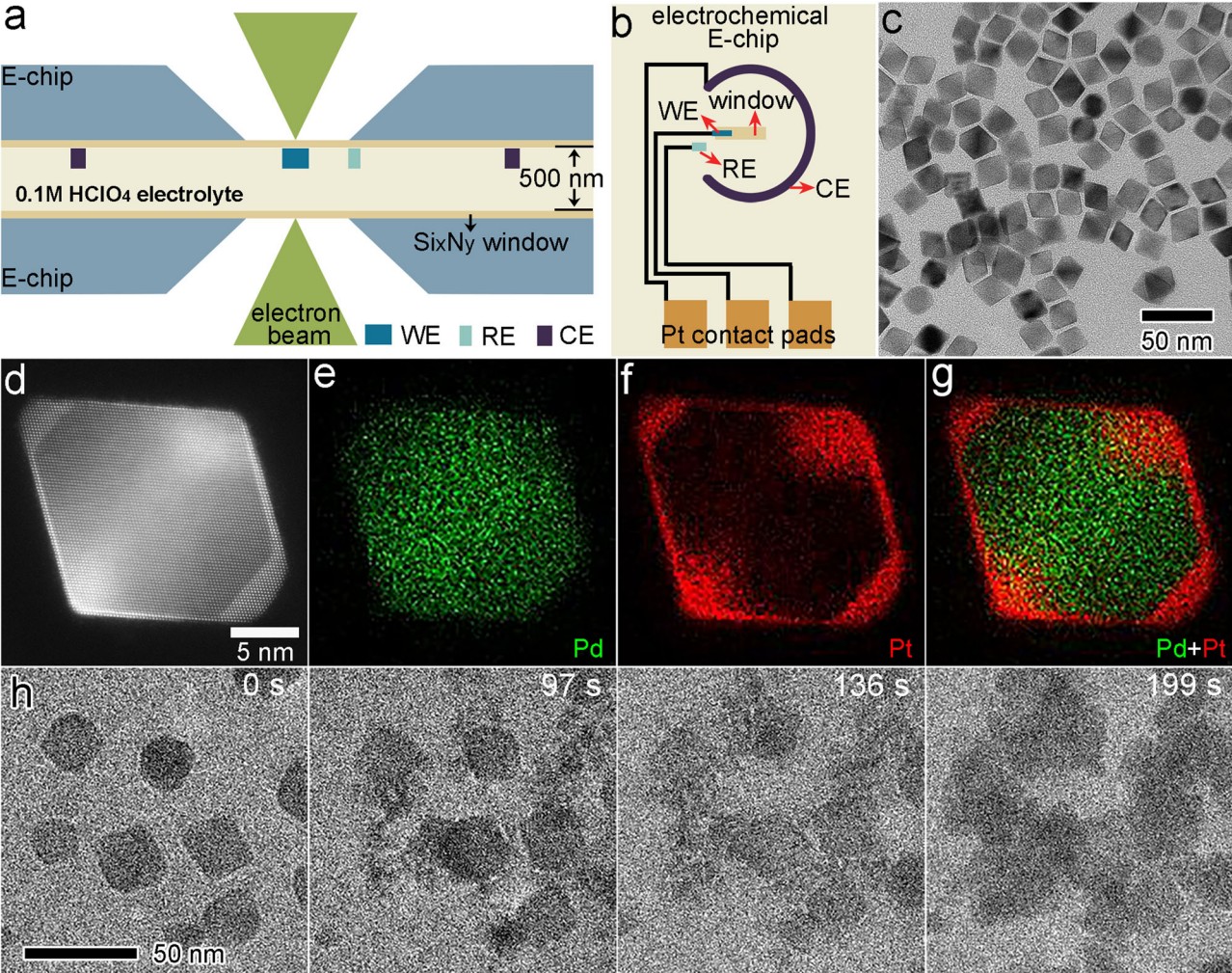

**Fig. 1 | In-situ electrochemical liquid cell technique and the rapid electrochemical corrosion of Pd@Pt core-shell octahedral nanoparticles. a** Schematic diagram of an electrochemical liquid cell assembled by two E-chips. **b** Schematic illustration of the top electrochemical E-chip, which integrates a three-electrode system. **c** TEM image of Pd@Pt core-shell octahedral nanoparticles on $Si_xN_y$ window. **d** Atomic-resolution STEM image of a Pd@Pt octahedron. **e–g** EDS mapping of a Pd@Pt octahedron. The green and red colors correspond to Pd and Pt elements, respectively. **h** Time-sequential in-situ TEM images of electrochemical corrosion process under the CV potential from −0.5 V to 0.5 V vs. Pt.

condition was recorded in Fig. 2c. During the reduction swapping (0 s–140 s), Island *a* exhibited rapid growth, particularly at the first minor reduction peak (-0.34 V- -0.38 V, in Fig. 3a). This growth can be attributed to the reduction of Pt ions and the subsequent deposition on the corners/edges of Island *a*[55]. Furthermore, a new island formed nearby (60 s) concurrently with the reduction Peak 1 (−0.39 V- −0.51 V, in Fig. 3a), indicating the formation of Pd island on the surface. Then, these two islands grew and coalesced (90 s). Additionally, more clusters nucleated on the corners (120 s, red arrows in Fig. 2c and Supplementary Fig. 7). In contrast, during the oxidation period (140 s–280 s), oxidative etching occurred preferentially at the top of the islands (green arrows in Fig. 2c). The preference can be attributed to the higher surface energy of the tip with a low coordination number (CN). As a result, the islands underwent etching and gradually disappeared (280 s). At the same time, a void appeared on the top-right surface (Void *b*), which differs from the preferential etching behavior observed at the corners in the available study of Pd nanocubes[5,56]. This discrepancy may arise from the stronger adsorption of hydroxide and oxygen species on Pt (100)/Pt (110) surfaces that are located at the corner/edge at high potentials[57], resulting in the preferential oxidation of Pt at the corner/edge. As a result, Pt oxides, which are undissolvable, cover the corner and thereby protect the internal metal atoms from

corrosion, as suggested by the Pourbaix diagram of Platinum[58,59]. This also reduces the surface energy at the corners and edges[60]. On the other hand, Pt on the terraces is more susceptible to oxidative etching due to the higher surface energy and surface incompleteness. Due to the strong adsorption, Pt atoms on the terraces would migrate to the corner/edge, leaving voids on the terrace. Consequently, the oxidative etching of both Pd and Pt is more likely to occur on the terraces of Pd@Pt core-shell octahedral nanoparticles at the beginning, leading to the formation of voids on the terraces.

We measured the area change of Island *a* and Void *b* (Fig. 2b) to gain insights into the morphological change of the electrocatalysts during the redox process (Fig. 2d). Notably, rapid area changes were observed in three regions (labeled by blue and red in Fig. 2d, e), which correspond to the redox peaks of the corresponding operando CV within TEM in Figs. 2f, 3a, respectively. Specifically, during the reduction period of Peak 1 (35 s–70 s), the area of Island *a* increased significantly (Fig. 2d). The corresponding area change rate in Fig. 2e shows a high growth rate (up to 0.50 nm²/s) of Island *a*, corresponding to the reduction of a small amount of Pt ions. Then, Pd ions were reduced and two islands formed in Fig. 2b, c (30 s–60 s). During the oxidation period of Peak 2 (200 s–215 s) and Peak 3 (225 s–250 s), the changes of Island *a* and Void *b* exhibited different behavior. Peak 2 at

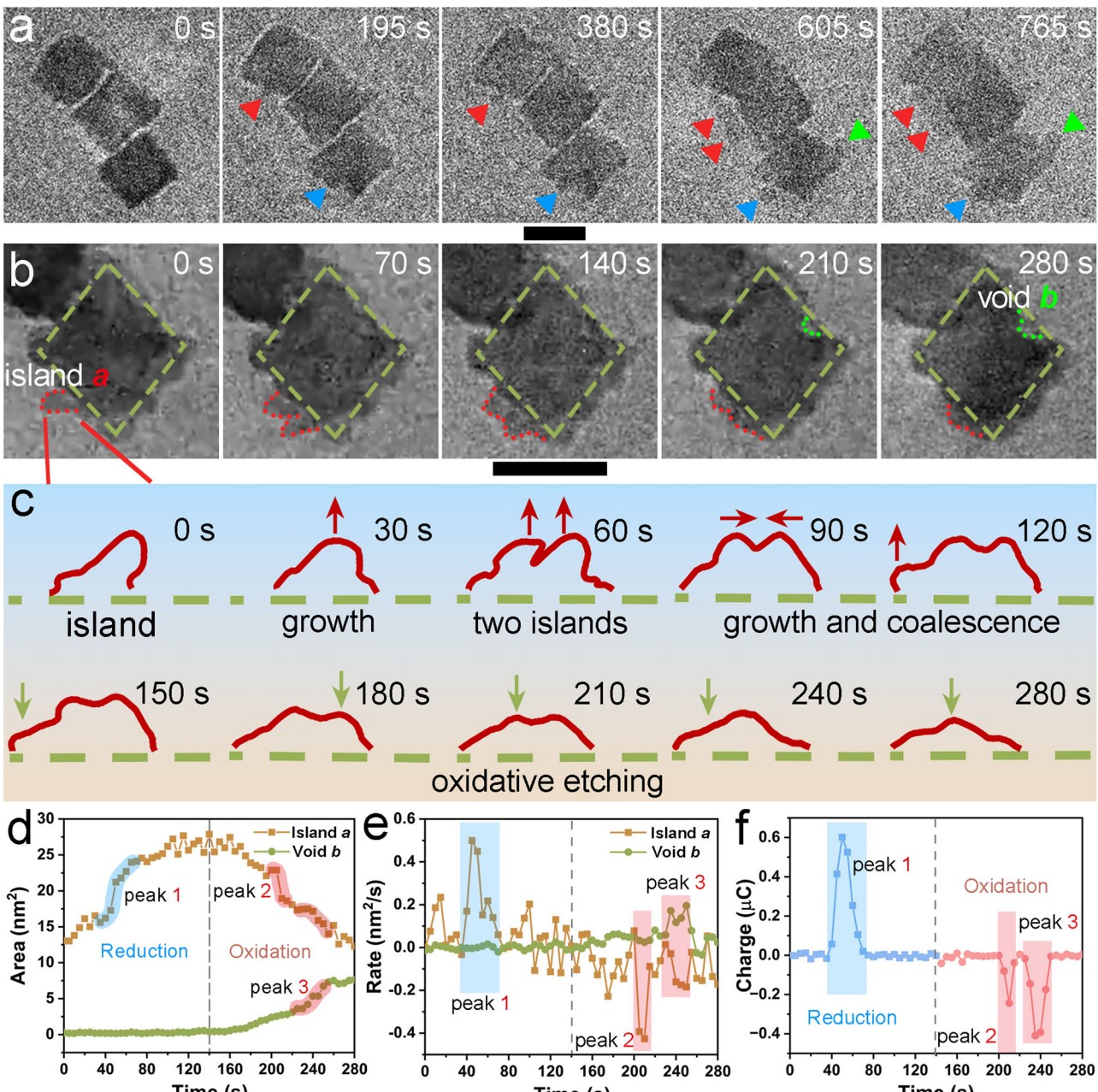

**Fig. 2 | Quantitative analysis of the electrochemical corrosion process using different CV ranges. a** Time-sequential in-situ TEM images of morphological evolution under the CV from −0.2 V to 0.2 V vs. Pt. **b** Time-sequential in-situ TEM images of morphological evolution under the CV from -0.9 V to −0.2 V vs. Pt. The red and green dotted areas are named Island *a* and Void *b*, respectively. **c** Trajectory analysis of Island *a* in (**b**). The trajectory diagram was obtained from the entire process shown in Supplementary Fig. 7. **d** Area changes of Island *a* and Void *b* in b as a function of time. **e** Corresponding change rate as a function of time. **f** Charge of the corresponding redox peaks calculated by integrating the peaks in the CV curve of Fig. 3a as a function of time. The scale bars in (**a**, **b**) are 20 nm.

−0.55 V corresponds to the oxidative etching of surface Pd, resulting in the fast etching rate (up to -0.43 nm²/s) of Island *a* (Fig. 2d, e), while Void *b* remained steady. At Peak 3 at −0.42 V, the etching rate of Island *a* decreased by approximately 50% (−0.17 nm²/s, Fig. 2e), while Void *b* grew much faster than that during Peak 2 (up to 0.20 nm²/s, Fig. 2e). This indicates a growing etching process on inner Pd at the higher potential. The potential difference of 0.13 V between Peak 2 and Peak 3 is close to the oxidation potential difference between Pd and Pt under the experimental conditions, according to the Pourbaix diagram. Therefore, Peak 3 corresponds to the oxidative etching of the surface Pt and subsequent exposure and oxidative etching of the inner Pd. Figure 2f shows the electric charge during the redox process

calculated by integrating the CV curve in Fig. 3a. The quantity of electric charge determines the etching rate of metal atoms. It was observed that the three regions with higher-charge in the operando CV curve (labeled by blue and red in Fig. 2f) correspond to the three regions with rapid area change within TEM (Fig. 2d, e). However, in Fig. 2e, the etching rate dominated by inner Pd atoms in Peak 3 is nearly equal to that of surface Pd atoms in Peak 2, despite the slightly larger total amount of electric charge during Peak 3 compared to Peak 2 in Fig. 2f. This difference may be attributed to the larger energy barriers caused by weaker adsorption of oxygen-containing species on inner Pd atoms, which hinders the oxidative etching process. Besides, the electric charge of the two oxidation peaks is almost equal to that of

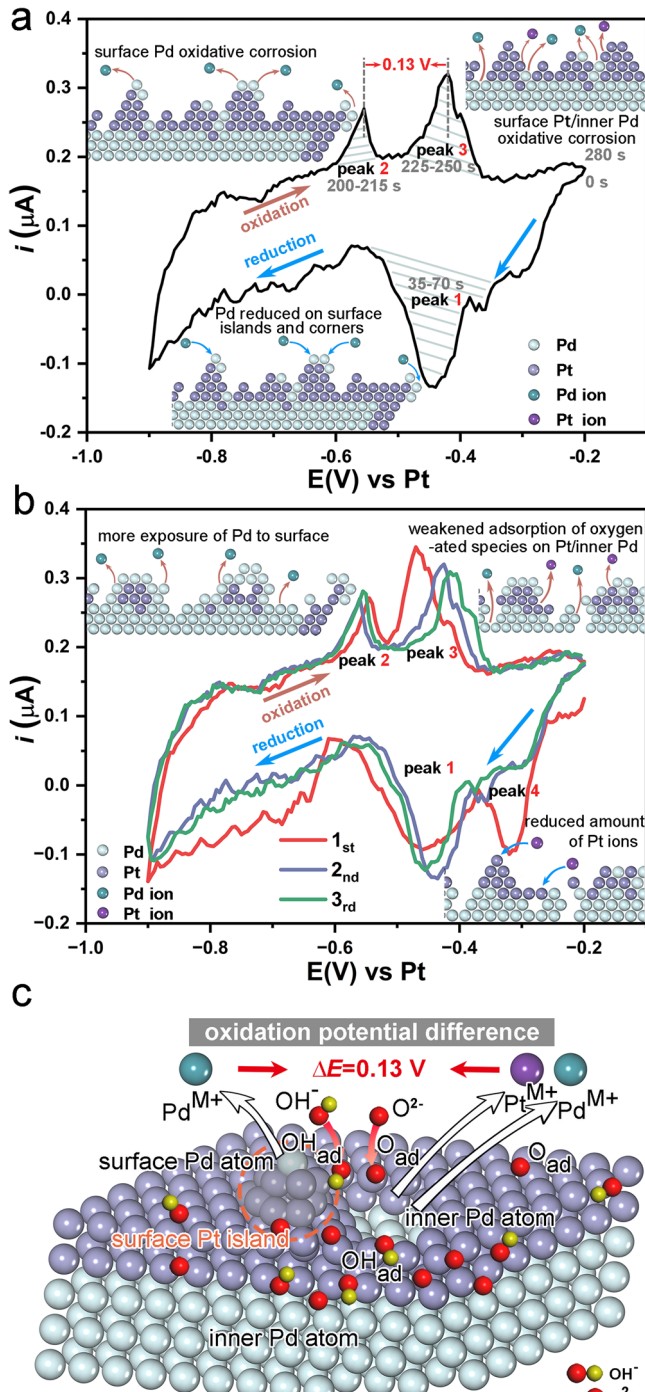

**Fig. 3 | CV analysis of the evolution process. a** The in-situ CV curve corresponds to the electrochemical process in Fig. 2c, and the illustrations of surface corrosion correspond to the redox peaks. **b** The adjacent CV curves showing the dynamic of the electrochemical corrosion process and the enlarged regions of the corresponding redox peaks. **c** Thematic illustration of the oxidation potential difference between surface Pd, Pt, and inner Pd atoms.

the reduction peaks, further demonstrating that those reduction and oxidation peaks mainly arise from the redox process of metallic catalysts. This highlights the precision of our in-situ electrochemical information.

In addition, we selected three adjacent CV curves to study the coherent evolutions of the redox process and the surface status of Pd@Pt core-shell octahedra (Fig. 3b). The three CV curves are marked in time sequence as cycle 1, 2, and 3, respectively. Considering that a

short period of CV from −0.2 V to 0.2 V (vs. Pt) was pre-applied, a certain amount of Pt atoms has been oxidized into the electrolyte. Thus, the corresponding Pt atoms were deposited on Pd@Pt nanoparticles corner/edge when swapping to the large reduction peak (Peak 4) starting from −0.28 V in cycle 1. However, a substantial weakening and a leftward shift of Peak 4 were observed in the two subsequent CV curves, indicating a decrease in the reduction rate of Pt atoms. This decrease can be attributed to the fewer Pt atoms available that participate in the cyclic redox at the lower operating voltage range (−0.9 V- −0.2 V) in the CV curves. During Peak 1, the reduction and deposition of Pd atoms on the corner/edge were the dominant process, exhibiting a relatively stable reduction potential. During the oxidation period, the onset potential of Peak 2 gradually shifted negatively, indicating the weak encapsulation of Pt atoms on the surface. With more Pd atoms exposed, they are more susceptible to corrosion. On the contrary, the onset potential of Peak 3 shifted positively, indicating the increasing difficulty in oxidizing the metal atoms. The continuous oxidative etching and reductive deposition led to the migration of more Pd atoms to the outer surface and Pt atoms toward the inner. Meanwhile, Pt atoms also moved to the corners and edges, weakening the adsorption of hydroxide and oxygen species on Pt atoms. Furthermore, Pt oxides formed on the particle surface as a result of the cyclic oxidation of Pt atoms, which gradually requires a higher potential to expose more inner Pd atoms. Thus, oxidative etching of those sites occurred at higher potentials.

In brief, the observed CV peaks can be attributed to specific processes occurring during the electrochemical redox reactions. Peak 1 corresponds to the reduction and deposition of Pd on the islands and edges. Peak 4 corresponds to the reduction and deposition of Pt on the corners/edges. Peak 2 represents surface Pd etching-dominated corrosion, and Peak 3 represents the inner Pd etching-dominated corrosion accompanied by a small amount of surface Pt corrosion. Interestingly, between Peak 2 and Peak 3, there is a noticeable potential gap of 0.13 V, which can be ascribed to the coverage of the surface layer of Pt atoms on the inner Pd atom. A higher potential is required to first oxidatively corrode the surface Pt atoms, and then the inner Pd atoms can be exposed and corroded. As shown in Fig. 3c, during the initial period of the redox reaction, the surface Pd on the islands can directly contact the electrolyte to adsorb oxygen-containing species more easily for oxidative etching. Then, with the corrosion of surface Pt atoms, Pd atoms were more completely exposed, inner Pd atoms can adsorb hydroxide and oxygen species and then were oxidized to form voids and further deposited on the corner/edge to form islands. With the cycling of the redox reaction, more Pd atoms were deposited on the islands and edges, and the oxidative etching of surface Pd became easier. Meanwhile, less exposure of Pt and the formation of Pt oxides led to higher Pt corrosion potential. Therefore, the structural dynamics of nanoparticles are closely correlated to the chemical changes in microenvironments related to the atomic sites and nanoparticle shape.

## Ex Situ Investigation on electrochemical corrosion of Pd@Pt octahedron electrocatalysts

Then, we performed an ex-situ electrochemical accelerated durability test (ADT) and characterized the catalysts after different ADT cycles (Fig. 4). The line scan of the outmost Pt layer shows the change of surface elements distribution of these particles after different cycles of ADT. In Fig. 4a–d, the selected nanoparticle before ADT has two intact Pt atomic layers, and the Pd and Pt elements were separated. After 10000 ADT cycles, the line scan (Fig. 4g) shows an increased strength of Pd signal, and a region (labeled by the red arrow in Fig. 4g, h) of Pt-Pd intermixing was present in the middle of the terrace sites, indicating that the Pd element migrated to Pt (111) surface during the CV cycles. During this process, several Pt surface steps formed (Supplementary Fig. 8), similar to the surface islands observed in the in-situ

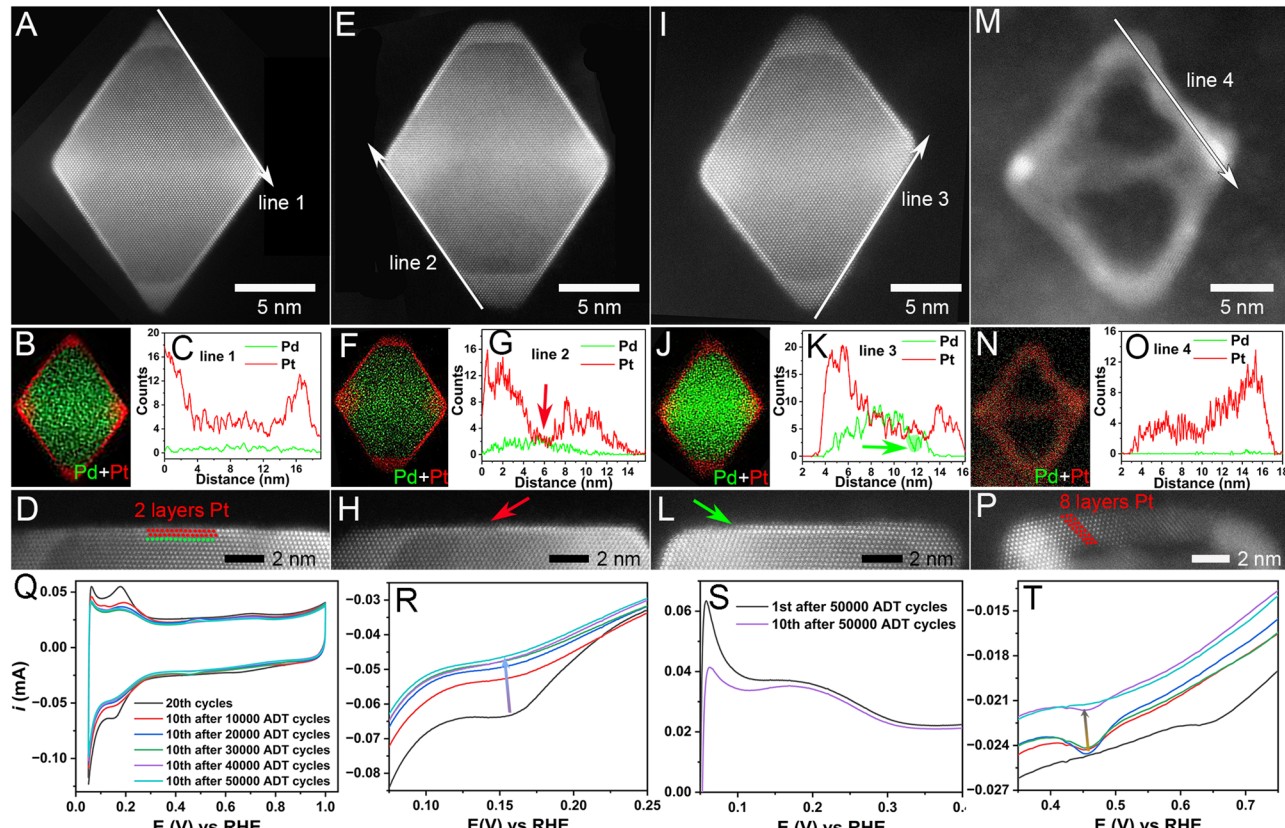

**Fig. 4 | HAADF-STEM characterization and CVs of Pd@Pt octahedron electrocatalysts after ex-situ electrochemical ADTs. a–p** Atomic-resolution HAADF-STEM images, EDS mapping, line scan, corresponding line scan area of Pd@Pt octahedra before ADT (**a–d**), after 10000 ADT cycles (**e–h**), after 30,000 ADT cycles (**i–l**), and after 50,000 ADT cycles (**m–p**), respectively. **q** Corresponding CVs before and after different ADT cycles. **r** Potential ranges from 0.075 V to 0.25 V vs. RHE in q. **s** 1st and 10th CVs after 50,000 ADT cycles with potential range from 0.05 V to 0.25 V vs. RHE. **t** Potential ranges from 0.35 V to 0.75 V vs. RHE in (**q**).

experiments caused by the oxygen adsorbate-driven migration of Pt atoms from {111} terraces to corners/edges. This could lead to the damage of partial surface Pt protection and the consequent corrosion of inner Pd atoms. After 30,000 ADT cycles, the Pd signal intensity was even stronger (Fig. 4k). Notably, there was a Pd signal loss (labeled by the green arrow in Fig. 4k), and the corresponding area is shown in Fig. 4l, indicating that the inner Pd atoms were etched away, which was further confirmed by the HAADF-STEM images of the same nanoparticle in a different orientation (Supplementary Fig. 9), showing the formation of a void. This phenomenon was similar to the formation of Void *b* during in-situ observation (Fig. 2b). Finally, the nanoparticles after 50,000 ADT cycles exhibited different degrees of nanoframe morphology (Supplementary Fig. 10). The complete nanoframe structure has only Pt, as indicated by both the EDS mapping (Fig. 4n) and line scan (Fig. 4o). The edge of the nanoframe (Fig. 4p) has eight layers of Pt atoms. It was much thicker than the Pt surface layers of the initial nanoparticles before ADT. This can be attributed to the adsorbate-driven reconstruction of Pt[61,62]. The high-indexed surface sites of Pt at the corners and edges exhibit stronger adsorption of hydroxide and oxygen at high potentials[57], leading to Pt atoms migrating to the corners and edges. In addition, Pt atoms at the corner/edge are preferentially oxidized to undissolved Pt oxide, covering the edge and corner to protect the internal metal atoms from corrosion.

The CVs before and after different ADT cycles are shown in Fig. 4q, showing the correlation between surface evolution and the change in adsorption of hydrogen and oxygen. The typical electrochemical CV curves show the conventional features associated with the adsorption and desorption of hydrogen and oxygen. We chose a part of the $H_2$ adsorption range (0.075 V to 0.25 V, Fig. 4r) to show the evolution of

$H_2$ adsorption region on the surface clearly. From the initial CV to that after 50000 ADT cycles, the area of $H_2$ adsorption continuously decreased, besides the weak contact of Pt nanoparticles with the carbon supports, the corrosion of carbon supports, and the loss of some surface atoms[11,35], this significant change in $H_2$ adsorption is largely attributed to the continuous corrosion of inner Pd atoms and the morphology collapse-induced decrease in surface active area. The comparison of the 1st with the 10th CV curves after 50000 ADT cycles indicated that $H_2$ continuously penetrated the surface Pd lattice in a short time[34], and the $H_2$ desorption became weak (Fig. 4s). Finally, with the inner Pd atoms being etched and the formation of Pt nanoframes, the $H_2$ adsorption area decreased notably after 50000 ADT cycles (Fig. 4r). Meanwhile, Fig. 4t shows the decreasing intensity of the reduction peak of Pd-O during 50000 ADT cycles. As fewer Pd atoms were left on the nanoparticle surface and the surface morphology collapsed, the corrosion of inner Pd atoms was slowed down, resulting in a decrease in the reduction peak of Pd-O after 50000 ADT cycles (Fig. 4t).

Based on the in-situ and correlative electrochemical measurement, the mechanism of electrochemical corrosion of Pd@Pt core-shell octahedra is summarized in Fig. 5. In the initial CV process (Stage 1), the surface reconstruction could be divided into three steps. First, oxidative species like oxygen adsorb on the Pt surface during the oxidation period. Then, the atomic exchange occurs between Pt atoms and oxide species during oxidation[15,63], which leads to the migration of surface Pt atoms and unsound Pt layers. With the process of CVs, atomic exchange moves internally to form the Pt surface steps or islands, and the inner Pd atoms are exposed to the electrolyte. From the ex-situ experiments, the surface evolution was not obvious; only

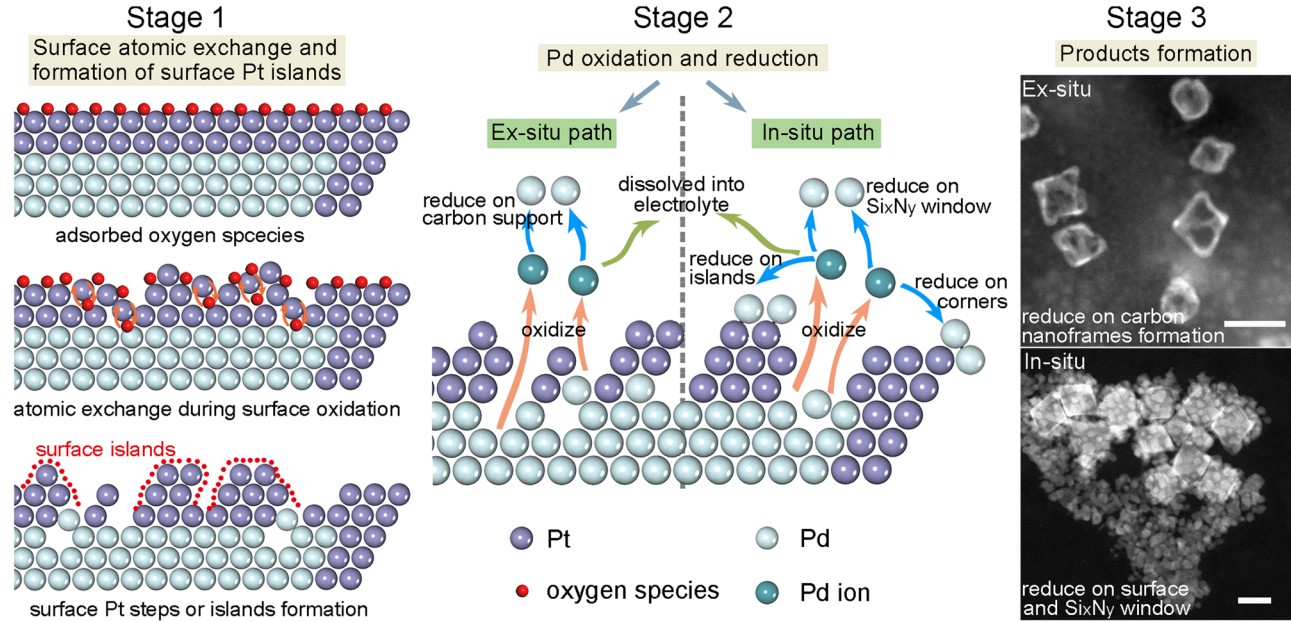

**Fig. 5 | Schematic diagram of the mechanisms of electrochemical corrosion via ex-situ and in-situ paths, respectively.** The electrochemical corrosion paths of ex-situ and in-situ include three stages: stage 1 is the surface atomic exchange and formation of surface Pt islands, stage 2 is the oxidation and reduction of Pd atoms, and stage 3 is the formation of different products. Scale bars: 20 nm.

surface steps[14] were generated (Supplementary Fig. 8), during the in-situ experiments, higher potentials led to faster reaction rates and the formation of surface islands (Fig. 2a). In Stage 2, the Pd atoms undergo oxidation and reduction with CVs, but the processes are slightly different due to the different environments in ex/in-situ experiments. During the ex-situ ADT process, inner Pd atoms are etched gradually, because the Pd@Pt nanoparticles were supported on carbon black (Vulcan® XC-72R) in the ex-situ experiments, most Pd ions are reduced and deposited onto carbon supports due to the large surface area of carbon sheets. For in-situ experiments, the continuous CVs cause the oxidative etching of surface Pd first, followed by the inner Pd atoms. During the reduction, most of the Pd ions are re-deposited onto the surface and edges of the metallic nanoparticle. The rest Pd ions are reduced onto the $Si_xN_y$ substrate (window of the liquid cell). Finally (Stage 3), Pt nanoframes form after long-term ex-situ ADT, and the carbon supports are decorated by Pd atoms (Fig. 5, Stage 3), confirmed by EDS mapping (Supplementary Fig. 11).

In conclusion, we performed in-situ ELC-TEM to investigate the nanoscale corrosion behavior of Pd@Pt core-shell octahedral nanoparticles during the electrochemical CV conditions. The reactions and dynamic process of the electrochemical corrosion are dependent on the applied CV potentials. Tracking the surface reconstruction in individual nanoparticles offers insight into the structural correlation with the redox process. Our in-situ experimental results show that the most stable (111) facets of Pt became unstable and more susceptible to corrosion even compared with the corners and edges of Pd@Pt core-shell octahedra in electrolyte during cyclic electric field swap, due to the change in surface affected by OH⁻ and $O^{2-}$ adsorbates. Connecting the visible structure changes with the in-situ electrochemical CV spectra reveals that the structure corrosion proceeds with the etching of inner Pd atoms, dissolution of Pt atoms during the oxidation and the redeposition of Pd and Pt onto the exterior surface during the reduction process in a CV cycles. Redeposition occurs preferentially at the corner/edge sites. Moreover, changes between the CV series are correlated with the chemical change in microenvironments related to the atomic sites and nanoparticle shape. Most of the intermediate process and time-resolved structure-property correlation during the electrochemical corrosion of core-shell catalysts cannot be observed by ex-situ study, but can be systematically analyzed through in-situ visualization, representing a big step forward towards understanding the fundaments of electrochemistry in catalysis.

## Methods

### Chemicals and materials
Sodium tetrachloropalladate(II) (Na₂PdCl₄, 99.99%), poly(-vinylpyrrolidone) (PVP, Mw ≈ 55,000), potassium chloride (KCl, 99.8%), potassium bromide (KBr, 99.9%), ascorbic acid (AA, 99%), chloroplatinic(IV) acid (H₂PtCl₆, 99.9%), concentrated perchloric acid (HClO₄, 70%), butylamine (98%) and glucose were purchased from Sigma-Aldrich. Oleylamine (OAm, 80%-90%) was purchased from Aladdin. Formaldehyde (HCHO, 37%-40%), methanol (99.9%), ethanol (99.8%), acetone (99.5%), toluene (99.5%), isopropanol (99.5%), chloroform (99.8%), and n-butylamine (99%) were purchased from Sinopharm Chemical Reagent Co., Ltd. Amorphous carbon (C, Vulcan® XC-72R) was purchased from Cabot. Nafion solution (5 wt.%) was purchased from DuPont, inc. Deionized water (18.2 MΩ·cm).

### Synthesis of 18 nm Pd@Pt octahedral nanoparticles
The typical synthesis process contained four steps: (1) synthesis of 6 nm Pd cubic seeds; (2) synthesis of 14 nm Pd octahedral seeds; (3) phase transfer of Pd seeds; (4) synthesis of Pd@Pt core-shell octahedral nanoparticles.

For step one (synthesis of 6 nm Pd cubic seeds), 8 ml of DI water containing 105 mg PVP, 60 mg AA, 5 mg KBr, 185 mg KCl was placed in a 20 ml vial and was pre-heated to 80 °C for 10 min. Then, 3 ml of DI water containing 57 mg Na₂PdCl₄ was added into the above solution using a pipette. The reaction was kept at 80 °C and under magnetic stirring (280 rpm) for 3 h. Finally, the products were washed with ethanol and acetone three times using a centrifuge and re-dispersed in ethanol for further use.

For step two (synthesis of 14 nm Pd octahedral seeds), 8 ml of DI water containing 105 mg PVP, 0.2 ml 6 nm Pd cubic seeds (washed from step one), 0.1 ml HCHO was placed in a 20 ml vial and was pre-heated to 60 °C for 5 min. Then, 3 ml of DI water containing 20 mg Na₂PdCl₄ was added into the above solution using a pipette. The reaction was kept at 60 °C and under magnetic stirring (280 rpm) for

3 h. Finally, the products were washed with ethanol and acetone three times using a centrifuge and re-dispersed in ethanol for further use.

For step three (phase transfer of Pd seeds), the 14 nm Pd octahedral seeds were dispersed in a mixed solution that contained 6 ml ethanol, 5 ml OAm, and 2 ml toluene. This solution was kept at 80 °C under a magnetic stirring (280 rpm) in a fume hood overnight. Then, the products were washed with OAm and ethanol three times using a centrifuge and re-dispersed in OAm for further use.

For step four (synthesis of Pd@Pt core-shell octahedral nanoparticles), 5 ml OAm contains 2.4 mg Pd octahedral seeds, and 30 mg glucose was pre-heated to 180 °C for 10 min. Then 0.265 ml OAm containing 2.65 mg $H_2PtCl_6$ was added into the above solution using a pipette. The reaction was kept at 180 °C and under magnetic stirring (280 rpm) for 3 h. Finally, the products were washed with chloroform and ethanol three times using a centrifuge and re-dispersed in chloroform for further use.

### Materials characterization

Transmission electron microscope (TEM) was conducted by the JEOL JEM-2100F at an accelerating voltage of 200 kV. TEM images of the intermediate products during the entire synthesis are shown in Supplementary Fig. 1. The high-angle annular dark-field scanning transmission electron microscopy (HAADF-STEM) and the energy dispersive spectra (EDS) elemental mapping were conducted by a JEOL ARM300FC aberration-corrected microscope at 300 kV (the facilities are in the Irvine Materials Research Institute (IMRI) at the University of California, Irvine) and a FEI Talox F200X G2 microscope at 200 kV (the facilities are in the Instrument Analysis Center at Shanghai Jiao Tong University). The model of the aberration corrector was CEOS GmbH. The size of the nanoparticles was defined by the edge length of the particles in TEM images. XRD data were recorded at a rate of 5°/min from 30° to 90° on a LabX XRD-6100. Inductively coupled plasma-atomic emission spectrometry (ICP-AES) data were achieved by an iCAP6300 instrument in the Instrument Analysis Center. The mass ratio of Pd and Pt was confirmed by ICP-AES in Table S1. The accurate mass ratio of Pd@Pt octahedral nanoparticles and the carbon support were achieved by thermogravimetric analysis (TGA, discovery TGA55) using oxygen gas with a flow rate of 60 ml/min at 500 °C for 40 min.

### In-situ characterizations

In-situ characterizations were conducted by a Poseidon Select (Protochips, USA) TEM holder. The experiments were carried out by a JEOL JEM2800 in IMRI. In a typical in-situ experiment (Supplementary Fig. 14), a large electrochemical E-chip (ETC-45CR-10) and a small E-chip (ECB-39A) were assembled as the electrochemical liquid cell. After sample and electrolyte loading, a Leak check was then carried out by a vacuum pump. After the successful pass of the leak check, the liquid cell TEM holder was loaded into the TEM. Next, an electrochemical workstation (Gamry, Reference 600, USA), together with a laptop, was connected to the holder. The TEM images recorded changes in the sample during the CV tests in the electrochemical liquid cell. The electrochemical cell hosts a bottom E-chip with a three-electrode system: a glassy carbon working electrode (WE), a Pt reference electrode (RE), and a Pt circular counter electrode (CE). The solution of 0.1 M $HClO_4$ (pH = 1 ± 0.1) was used as the electrolyte for all the measurements, which was diluted from 70% double-distilled perchloric acid with DI water when it was used for freshness. Before the in-situ electrochemical experiments, we adjusted the beam dose rate (50 ~ 60 e/Å²·s) to ensure the electron beam would not damage the materials (Supplementary Fig. 12). To calibrate the Pt reference electrode, the same normal CV curves of commercial Pt/C catalysts were measured using the reversible hydrogen electrode (RHE) and Pt wire as reference electrodes (Supplementary Fig. 13). The CV conditions we used in the experiments were described as follows: (1) from −0.5 V to 0.5 V, 10 mV/s; (2) from −0.2 V to 0.2 V, 5 mV/s; (3) from −0.9 V to

−0.2 V, 5 mV/s. The video acquisition was conducted by a OneView camera (Gatan, USA). Video processing and area measurement were carried out using VirtualDub and Image J software.

### Preparation of electrocatalysts

The as-prepared Pd@Pt octahedral nanoparticles were first dispersed in chloroform and added into a chloroform solution containing carbon black (Vulcan® XC-72R) and sonicated for 1 h. The mass ratio of nanoparticles and carbon was 1:4. After that, the mixture was sonicated for an additional 30 min and then stirred overnight. The products were collected by a centrifuge and re-dispersed in n-butylamine at a concentration of 2 mg-catalysts/ml. Next, the solution was stirred for 3 days to remove the organic compound. After that, the products were washed with methanol and collected by a centrifuge. Freeze drying was then conducted to achieve dry electrocatalysts. Finally, 30 min low temperature (250 °C) thermal annealing was carried out using a high-temperature furnace (OTF-1200X-S, Kejing, Hefei) to further remove the surface organic compound. The mass ratio of Pd@Pt nanoparticles and carbon was confirmed by TGA in Table S1.

### Ex-situ electrochemical measurements

The electrochemical ink was prepared by dispersing the as-prepared electrocatalysts into a mixed solution of 1 ml isopropanol, 4 ml DI water, and 25 μl Nafion solution. After being sonicated for 15 min, the ink was prepared to be loaded in the working electrode (WE). The electrochemical tests were conducted by a three-electrode system that contained a working electrode (a glassy-carbon rotating disk electrode, RDE) from Pine Instruments with an area of 0.196 cm², a reference electrode (RE, a HydroFlex hydrogen electrode), and a counter electrode (CE, a platinum wire). Calibration of RE was conducted before the test via hydrogen evolution reaction (HER) with $H_2$ oxidation/evolution on RE, and all the potentials we applied in ex-situ tests were referenced to reversible hydrogen electrode (RHE). 40 μl ink was loaded on the WE to do the electrochemical test. Similar to in-situ experiments, 0.1 M $HClO_4$ solution was used as the electrolyte. CV test was operated from 0.05 V to 1 V at a scan rate of 50 mV/s in the argon (Ar)-saturated electrolyte. The accelerated durability test (ADT) was conducted from 0.6 V to 1 V at a scan rate of 100 mV/s. CV tests were carried out before and after different cycles of ADT, respectively. The CVs recorded in Fig. 4 were the 20th and the 10th cycles after 19 and 9 cycles of stabilization before and after ADT, respectively. All electrochemical data were not iR corrected.

### Data availability

The data that support the findings of this study are available from the corresponding authors J.W. upon request.

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

## Acknowledgements

This work was sponsored by National Key R&D Program of China (No. 2017YFB0406000), the National Science Foundation of China (21875137, 51521004, and 51420105009), Program of Shanghai Academic/Technology Research Leader (Project No. 23XD1422100), and the Innovation Program of Shanghai Municipal Education Commission (Project No. 2019-01-07-00-02-E00069), the 111 Project (Project No. B16032) (J.W.), and the support from Center of Hydrogen Science and Joint Research Center for Clean Energy Materials from Shanghai Jiao Tong University. We also acknowledge the Instrument Analysis Center at Shanghai Jiao Tong University for partial characterizations of ICP and EDS mapping.

## Author contributions

F.S., H.H., W.G., J.W. wrote the manuscript. F.S., J.W. conceived and designed the study. F.S., P.T. (Peter Tieu), and W.G. performed the in-situ experiments. P.T. (Peter Tieu), F.S., and F.L. performed the STEM and EDS work. F.S., J.P., and H.H. analyzed the in-situ data. W.Z. helped with the ex-situ electrochemical test. P.T. (Peter Tieu), W.S., P.T. (Peng Tao), C.S., and T.D. helped with data analysis. W.G. and X.P. revised the manuscript.

## Competing interests

The authors declare no competing interests.
