## [Peer Review File · Nature Communications]

Direct In-Situ Imaging of Electrochemical Corrosion of Pd-Pt Core-Shell ElectrocatalystsREVIEWER COMMENTS

Reviewer #1 (Remarks to the Author):

In the submitted manuscript, the authors have undertaken a comprehensive study on the nanoscale structural evolution of Pd@Pt core-shell octahedral nanoparticle electrocatalysts using in-situ electrochemical liquid cell transmission electron microscopy (ELC-TEM). This is a significant and compelling topic, as understanding the corrosion behaviors in the microenvironment during electrocatalysis is crucial for designing more stable catalysts, particularly those based on Platinum (Pt), which are indispensable in various energy conversion processes. The authors have successfully utilized the ELC-TEM technique to monitor the real-time structural evolution of the Pd@Pt electrocatalyst during a complete cyclic voltammetry (CV) cycle. The findings provide valuable insights into the surface reconstruction involving preferential oxidative etching and reductive re-deposition of Pd and Pt atoms, driven by hydroxide/oxygen adsorbates. The correlation of these structural dynamics with the electrocatalyst's performance is a significant contribution to the field.

The manuscript is well-structured, and the narrative is logical and easy to follow. However, I recommend some revisions for the authors to consider for improving the clarity and impact of the manuscript.

(1) The authors have done an admirable job explaining both their experimental procedure and results. However, further elaboration is needed for certain figures, particularly Fig 4q. It would be helpful if they provided a more in-depth interpretation of these figures to enhance readers' comprehension of their results.

(2) The actual electrocatalytic setup and the electrolytic cell used for in situ liquid transmission electron microscopy differ significantly in many aspects, such as the concentration of the electrolyte, the use of the reference electrode, and the electrochemical procedure. The authors should discuss how these differences affect their experimental results to allow the reader to clarify the applicability of the study's conclusions.

(3) One of the main findings of this study is that the corrosion behavior of Pd@Pt core-shell octahedral nanoparticles under electrochemical CV conditions is related to the applied CV potential. However, actual electrocatalytic devices may operate over a wider range of potentials, which may affect the kinetics and mechanisms of corrosion. The authors should discuss this possibility and examine the corrosion behavior over a wider range of potentials.

(4) The paper indicates an increase in Pd signal intensity after 10000 ADT cycles, which further strengthens after 30000 ADT cycles. However, it would be advantageous if the authors could provide a more quantitative analysis of this augmentation in Pd signal intensity. This could be accomplished by presenting numerical values or conducting statistical analysis to substantiate these observations.

(5) The conclusion section offers a comprehensive summary of the findings. Nevertheless, it would be beneficial if the authors could discuss the implications of these findings within a broader context. For instance, how does the corrosion of Pd@Pt core-shell octahedra affect their performance as electrocatalysts? What potential strategies can be employed to mitigate this corrosion? Furthermore, it would be intriguing to see a more detailed discussion on how these findings can guide the development of more stable and efficient Pt-based electrocatalysts.

(6) Some related literatures regarding electrochemical liquid cell TEM can be cited, such as, Nat. Protoc. 2023, 18, 555-578; Acc. Chem. Res. 2017, 50, 8, 1808-1817; Science 2015, 350, aaa9886.

(7) Some typos should be corrected. For example: supplementary Fig. 9b "showing he etching region of the inner Pd atoms" in manuscript. Authors should correct other similar mistakes.

Overall, this study provides a significant contribution to the understanding of the structural evolution of Pt-based electrocatalysts under operating conditions. I recommend its publication following the suggested revisions.

Reviewer #2 (Remarks to the Author):

This paper illustrates the use of and in situ electrochemical liquid cell (ELC) for the study of Pt-Pd nanoparticles and their changes during catalysis. These are then compared to ex situ observations. As illustrated in Fig 5, the nanoparticles in the in situ and ex situ experiments undergo different processes and so this does mean that the in situ experiments have less practical value. As mentioned by the authors the main value of this work is of a more fundamental nature.

General comments

1) Although kinetics are mentioned throughout the paper no kinetics are calculated and there is no detailed discussion of kinetics.

For example, in the conclusion "The reactions and kinetics of the electrochemical corrosion are dependent on the applied CV potentials"

2) A more in depth discussion of why in situ and ex situ experiments give different results would be useful. Is this due to the cell setup? Transport within the cell?

3) The processes observed in the in situ study are not especially new and known processes. Highlighting the novelty of the observations would help.

Reviewer #3 (Remarks to the Author):

In this study, the authors employed in-situ liquid cell TEM to observe the electrochemical corrosion behaviors of Pd@Pt nanoparticles. The in-situ TEM images provide some information during the corrosion process that are difficult to capture using ex-situ methods, but at a significantly reduced resolution. The proposed mechanism lacks sufficient support from the data and evidence presented. Furthermore, the significance and novelty of this process/mechanism do not appear to be groundbreaking enough to warrant publication in Nature Communications. I suggest considering an alternative journal for publication, subject to the clarification of several key questions.

1. Due to its low resolution of the Supplementary Fig. 4, it will be more convincing if it has similar magnification and resolution as those in Figs. 1d-g, to confirm that the core-shell structure indeed transformed to PdPt alloy nanoparticles.

2. In the oxidation period, e.g., stage 1, if the corrosion process follows the proposed mechanism, there should be Pt oxides forming at the corner/edge, which can be and need to be confirmed by experimental characterizations, for example, the lattice distance in the images, either the in-situ or ex-situ images, or EDS and EELS, etc. In the manuscript, no oxides are evidenced in the experimental data, making the proposed corrosion process and mechanism less convincing.

3. In Fig. 2a (605 seconds), the "new island" indicated by red triangles might not originate from the corrosion of the highlighted particle. Instead, it appears to be from a particle moving in from the lower left corner, as observed in the movie. The image below, captured from the movie at 9:50, shows the particle from the lower left corner approaching the central crystal, with the "island" located in between marked by red arrows. Tracking the formation of the "island" throughout the entire movie suggests that it is unlikely to have derived from the central crystal.

4. Supplementary Fig. 12 shows obvious crystal morphology evolution (indicated by red circles), resembling the change in crystal morphology observed in Figure 2 under electrochemical conditions. This finding implies that the electron beam could play a role in altering the crystal morphology.

Fig.2

Supplementary Fig. 12 Time-sequential in-situ TEM images of Pd@Pt octahedral nanoparticles without electrochemical conditions showed that the electron beam didn't damage the materials. The scale bar was 50 nm. The dose rate we used in all the in-situ experiments was $50 \sim 60 \text{ e}/\text{\AA}^2 \cdot \text{s}$.

5. In Fig. 3, it is suggested to use visually different colors to represent Pd, Pt, Pd ion, and Pt ion for a clearer illustration.

6. Please correct typos throughout the manuscript. For example, in Page 3, "combination with aberration and ..." should be "combination with aberration-correction and ..."; In Supplementary Fig. 9, "green circle" in the legend should be "green rectangle".

Reviewer #4 (Remarks to the Author):

The work by Wu and co-workers used in situ electrochemical liquid-cell TEM to track the morphological change during oxidative corrosion of Pd@Pt nanoparticles. Overall the experiments well performed with decent quality of the electrochemical data acquired in liquid-cell TEM. The reviewer points out the following critical questions should be addressed before this work can be considered for publication in Nature Commun.

1. Interpretation of CV profiles in Fig.3, such as assignment of two oxidative peaks, require reference data of CV profiles of Pd@Pt core-shell NPs in standard 3-electrode electrochemical cell.
2. The potential of Pt pseudo-RE in 0.1M acid needs calibration against reversible hydrogen electrode (RHE). It is very important to know the potential of Pd/PdOx and Pt/PtOx on the RHE scale and compare to literature reports in electrochemistry/catalysis community.
3. The reviewer found that analysis of contrast based on TEM images (Fig. 2) is limited to extract reliable information. STEM imaging in liquid is highly recommended to directly show the etching of Pd vs. Pt as those two elements have very different Z contrast.

Reviewer #1:

In the submitted manuscript, the authors have undertaken a comprehensive study on the nanoscale structural evolution of Pd@Pt core-shell octahedral nanoparticle electrocatalysts using in-situ electrochemical liquid cell transmission electron microscopy (ELC-TEM). This is a significant and compelling topic, as understanding the corrosion behaviors in the microenvironment during electrocatalysis is crucial for designing more stable catalysts, particularly those based on Platinum (Pt), which are indispensable in various energy conversion processes. The authors have successfully utilized the ELC-TEM technique to monitor the real-time structural evolution of the Pd@Pt electrocatalyst during a complete cyclic voltammetry (CV) cycle. The findings provide valuable insights into the surface reconstruction involving preferential oxidative etching and reductive re-deposition of Pd and Pt atoms, driven by hydroxide/oxygen adsorbates. The correlation of these structural dynamics with the electrocatalyst's performance is a significant contribution to the field.

The manuscript is well-structured, and the narrative is logical and easy to follow. However, I recommend some revisions for the authors to consider for improving the clarity and impact of the manuscript.

Overall, this study provides a significant contribution to the understanding of the structural evolution of Pt-based electrocatalysts under operating conditions. I recommend its publication following the suggested revisions.

Response:

We would like to thank the reviewer for the positive and valuable comments. Based on the suggestions, we provide the point-to-point responses here.

Comment #1: *"The authors have done an admirable job explaining both their experimental procedure and results. However, further elaboration is needed for certain figures, particularly Fig 4q. It would be helpful if they provided a more in-depth interpretation of these figures to enhance readers' comprehension of their results."*

Response: Thanks for the valuable suggestion. Fig. 4r and 4t are zoom-in parts of Fig. 4q, showing more details for us to discuss the surface evolution during the electrochemical CV process. We provide a more detailed interpretation for Figs 4q-4t. The revised content is highlighted in yellow in the revised manuscript and listed below:

On Page 17, Para 2, line 1: *"CVs before and after the different ADT cycles are shown in Fig. 4q and can help to correlate the surface evolution with the adsorption change of hydrogen and oxygen. besides the weak contact of Pt nanoparticles with the carbon supports and the corrosion of carbon supports^{58, 59}, As fewer Pd atoms were left and the surface morphology collapsed,"* has been revised to *"The CVs before and after different ADT cycles are shown in Fig. 4q, showing the correlation between surface evolution and the change in adsorption of hydrogen and oxygen. The typical electrochemical CV curves show the conventional features associated with the adsorption and desorption of hydrogen and oxygen. We chose a part of the H₂ adsorption range (0.075 V to 0.25 V, Fig. 4r) to show the*

evolution of H_2 adsorption region on the surface clearly. From the initial CV to that after 50000 ADT cycles, the area of H_2 adsorption continuously decreased, besides the weak contact of Pt nanoparticles with the carbon supports, the corrosion of carbon supports, and the loss of some surface atoms^{64, 65,}As fewer Pd atoms were left on the nanoparticle surface and the surface morphology collapsed,"

Comment #2: "The actual electrocatalytic setup and the electrolytic cell used for in situ liquid transmission electron microscopy differ significantly in many aspects, such as the concentration of the electrolyte, the use of the reference electrode, and the electrochemical procedure. The authors should discuss how these differences affect their experimental results to allow the reader to clarify the applicability of the study's conclusions."

Response: Thanks for the suggestion. All the experimental set-up of the in-situ and ex-situ experiments is in the Methods section, including the electrolyte, electrode system, and detailed electrochemical program setup. We also added a discussion about how these differences affect the experimental results. In supplementary materials, we added the data for the calibration of Pt wire against the RHE reference electrodes, explaining how to compare the results from the in-situ and ex-situ experiments. In the analysis of Fig. 5, we have discussed the difference in electrochemical corrosion processes through in-situ and ex-situ conditions, mainly due to a small number of different experimental conditions. The revised content is highlighted in yellow in the revised manuscript and listed below:

On Page 27, Para 2, line 7: "The CV tests were carried out simultaneously with the TEM movie recording." has been revised to "The TEM images recorded changes in the sample during the CV tests in the electrochemical liquid cell. The electrochemical cell hosts a bottom E-chip with a three-electrode system: a glassy carbon working electrode (WE), a Pt reference electrode (RE), and a Pt circular counter electrode (CE). The solution of 0.1 M $HClO_4$ was used as the electrolyte. To calibrate the Pt reference electrode, the same normal CV curves of commercial Pt/C catalysts were measured using the reversible hydrogen electrode (RHE) and Pt wire as reference electrodes. (Supplementary Fig. 13)."

Supplementary Fig. 13 has been added:

Supplementary Fig. 13 Calibration of Pt wire reference electrode using RHE.

Comment #3: "One of the main findings of this study is that the corrosion behavior of Pd@Pt core-shell octahedral nanoparticles under electrochemical CV conditions is related to the applied CV potential. However, actual electrocatalytic devices may operate over a wider range of potentials, which may affect the kinetics and mechanisms of corrosion. The authors should discuss this possibility examine the corrosion behavior over a wider range of potentials."

Response: We agree with the reviewer that different operating potential ranges will affect the kinetics and mechanisms of corrosion. Therefore, to optimize the experimental set-up for the observation the reaction process, we performed CV using different potential ranges, including from -0.5 V to 0.5 V vs. Pt in Fig. 1h, from -0.2 V to 0.2 V vs. Pt in Fig. 2a, and from -0.9 V to -0.2 V vs. Pt in Fig. 2b. According to the results, we found that a lower potential leads to slower corrosion rate, which allow us to clearly observe and quantitatively analyze the corrosion process.

Comment #4: "The paper indicates an increase in Pd signal intensity after 10000 ADT cycles, which further strengthens after 30000 ADT cycles. However, it would be advantageous if the authors could provide a more quantitative analysis of this augmentation in Pd signal intensity. This could be accomplished by presenting numerical values or conducting statistical analysis to substantiate these observations."

Response: Thanks for the reviewer's suggestion. We analyzed the line-scan region of EDX data in Fig. 4 and quantified the atomic contents of Pt and Pd. Results show the increase in Pd signal intensity after 10000 and 30000 ADT cycles (Fig R1).

Fig R1. Percentage of the elements Pt and Pd atoms in the line-scan regions in Fig. 4 after the different ADT cycles.

Comment #5: "The conclusion section offers a comprehensive summary of the findings. Nevertheless, it would be beneficial if the authors could discuss the implications of these findings within a broader context. For instance, how does the corrosion of Pd@Pt core-shell octahedra affect their performance as electrocatalysts? What potential strategies can be employed to mitigate this corrosion? Furthermore, it would be intriguing to see a more

detailed discussion on how these findings can guide the development of more stable and efficient Pt-based electrocatalysts."

Response: Thanks for the reviewer's valuable suggestion. In this work, we mainly focused on investigating the dynamics and mechanism of the electrochemical corrosion process of electrocatalysts during electrocatalysis to advance the understanding of the fundamentals of electrochemistry and electrocatalyst evolution. The dynamics of the intermediate states that are usually missing in ex-situ studies are key to revealing the dynamics of not only the catalyst degradation but also the formation of those highly active nanostructures. According to the in situ studies, the effect of the Pd@Pt core-shell octahedra corrosion on their electrocatalytic performance has been discussed in the manuscript, and the potential strategies that could mitigate corrosion have been proposed in previous articles by our group (*Nat. Commun.* **2018**, 9, 1011; *Adv. Mater.* **2021**, 33, 2101511; *Chem* **2020**, 6, 2257-2271): by selectively growing thicker Pt layers on the corners, (*Nat. Commun.* **2018**, 9, 1011; *Adv. Mater.* **2021**, 33, 2101511) and limit the size of the Pd@Pt nanoparticles, we can improve the catalyst durability in both an in-situ LC study and an ex-situ ORR stability test (*Chem* **2020**, 6, 2257-2271). In the electrochemical test in this work, we observed that the (111) terraces of the octahedron were more susceptible to corrosion, protecting the (111) surface by selectively growing thicker layers of Pt atoms on the surface could slow down the electrochemical corrosion. However, the effect on its performance still requires to be studied. In addition, we are also working on how to guide the design of more stable and efficient electrocatalysts through these findings of in-situ electrochemical corrosion experiments.

Comment #6: *"Some related literatures regarding electrochemical liquid cell TEM can be cited, such as, Nat. Protoc. 2023, 18, 555-578; Acc. Chem. Res. 2017, 50, 8, 1808-1817; Science 2015, 350, aaa9886."*

Response: Thanks for the reviewer's valuable suggestion. We have carefully read and cited the articles as you suggested. The revised content is highlighted in yellow in the revised manuscript and listed below:

On Page 4, Para 2, line 1: *" With the rapid development of in-situ liquid cells in TEM18-20, reactions in liquid environments can be recorded with high spatial resolution21-25" has been revised to " With the rapid development of in-situ liquid cells in TEM18-21, reactions in liquid environments can be recorded with high spatial resolution22-28."*

On Page 4, Para 2, line 11: *" By coupling the in-situ liquid cell TEM (LC-TEM)39 with an electrochemical workstation, the real-time structure evolution of catalysts during electrocatalysis can be monitored40-42." has been revised to " By coupling the in-situ liquid cell TEM (LC-TEM)39 with an electrochemical workstation40, the real-time structure evolution of catalysts during electrocatalysis can be monitored41-44."*

[21] Yang, R. et al. Fabrication of liquid cell for in situ transmission electron microscopy of electrochemical processes. Nat. Protoc. 18, 555-578 (2023).

[27] Ross, F. M. Opportunities and challenges in liquid cell electron microscopy. Science 350, aaa9886 (2015).

[28] Zeng, Z., Zheng, W. & Zheng, H. et al. Visualization of Colloidal Nanocrystal Formation and Electrode–Electrolyte Interfaces in Liquids Using TEM. Acc. Chem. Res. 50, 1808-1817 (2015).

[40] Zhang, Q. et al. In situ TEM visualization of LiF nanosheet formation on the cathode-electrolyte interphase (CEI) in liquid-electrolyte lithium-ion batteries. Matter 5, 1235-1250 (2022).

[44] Yang, R. et al. Fabrication of liquid cell for in situ transmission electron microscopy of electrochemical processes. Nat. Protoc. 18, 555-578 (2023).

Comment #7: *“Some typos should be corrected. For example: supplementary Fig. 9b “showing he etching region of the inner Pd atoms” in manuscript. Authors should correct other similar mistakes.”*

Response: Thanks for the reviewer's suggestion. We checked and corrected similar issues throughout the entire manuscript. The revised content is highlighted in yellow in the revised manuscript and listed below:

On Page 3, Para 1, line 14: *“ In combination with aberration and” has been revised to “In combination with aberration correction and”*

On Page 28, Para 2, line 7: *“ Celebration of RE” has been revised to “Calibration of RE”*

In Supplementary Fig. 9: *“ Fig. 9b, showing the etching region of he inner Pd atoms (green circle).” has been revised to “Fig. 9b, showing the etching region of the inner Pd (green rectangle).”*

Reviewer #2:

This paper illustrates the use of and in situ electrochemical liquid cell (ELC) for the study of Pt-Pd nanoparticles and their changes during catalysis. These are then compared to ex situ observations. As illustrated in Fig 5, the nanoparticles in the in situ and ex situ experiments undergo different processes and so this does mean that the in situ experiments have less practical value. As mentioned by the authors the main value of this work is of a more fundamental nature..

Response:

Thanks for the reviewer's valuable suggestions. We have modified our manuscript, and we provide the point-to-point response here. The changes in the manuscript are highlighted in yellow.

Comment #1: *"Although kinetics are mentioned throughout the paper no kinetics are calculated and there is no detailed discussion of kinetics.*

For example, in the conclusion "The reactions and kinetics of the electrochemical corrosion are dependent on the applied CV potentials"

Response: Thanks for the reviewer's suggestion. In this paper, we mainly focused on the dynamic process of electrochemical corrosion rather than the kinetics of electrochemical reactions. We have revised the description, and the revision is highlighted in yellow in the manuscript and listed below:

On Page 4, Para 1, line 2: *" It is essential to record the evolution of real catalysts using appropriate and advanced in-situ/operando technologies¹⁴⁻¹⁷, which can help researchers understand the kinetics of electrochemical corrosion more fundamentally and further guide material design based on corrosion mechanism studies."* has been revised to *"It is essential to record the evolution of catalysts under realistic operation conditions using appropriate and advanced in-situ/operando technologies¹⁴⁻¹⁷ to better understand the dynamic process in electrochemical corrosion."*

On Page 8, Para 1, line 1: *" to slow down the corrosion kinetics to clearly observe the beginning of the etching process."* has been revised to *" With the lower potential applied, the slower corrosion rate allows for clear observation of the etching process."*

On Page 10, Para 1, line 12: *" To further study the corrosion kinetics of the Pd@Pt ..."* has been revised to *"To further study the corrosion dynamics of the Pd@Pt ..."*

On Page 20, Para 2, line 3: *" The reactions and kinetics of the electrochemical corrosion are dependent on the applied CV potentials."* has been revised to *"The reactions and dynamic process of the electrochemical corrosion are dependent on the applied CV potentials."*

Comment #2: *"A more in depth discussion of why in situ and ex situ experiments give different results would be useful. Is this due to the cell setup? Transport within the cell?"*

Response: Thanks for the reviewer's valuable suggestion. In the analysis section of Fig. 5, we have discussed the different electrochemical corrosion processes between in-situ and ex-situ paths, mainly due to a number of different experimental conditions. First, the unsupported Pd@Pt nanoparticles were used in the in-situ experiments, while carbon-supported Pd@Pt nanoparticles were used in the ex-situ experiments; the support could influence the location of the Pd deposition when it is reduced. Then, the potential ranges applied in the in-situ and ex-situ experiments are different, which can lead to different Pd corrosion rates. However, the fundamental mechanisms of catalyst corrosion under electrochemical conditions are similar. We also added a detailed discussion to explain how to understand the results from in-situ and ex-situ experiments. The revised content is highlighted in yellow in the revised manuscript and listed below:

On Page 19, Para 1, line 9: "*only surface steps¹⁴ are generated (Supplementary Fig. 8), while during the in-situ experiments, due to higher potential, surface islands form (Fig. 2a).*" has been revised to "*during the in-situ experiments, higher potentials led to faster reaction rates and the formation of surface islands (Fig. 2a).*"

On Page 19, Para 1, line 13: "*inner Pd atoms are etched gradually, and most Pd ions are reduced onto carbon*" has been revised to "*inner Pd atoms are etched gradually, because the Pd@Pt nanoparticles were supported on carbon black (Vulcan® XC-72R) in the ex-situ experiments, most Pd ions are reduced and deposited onto carbon*"

Comment #3: "*The processes observed in the in situ study are not especially new and known processes. Highlighting the novelty of the observations would help.*"

Response: Thanks for the reviewer's valuable suggestion. To date, most research works focused on the structural evolutions like coarsening, selective dissolution, and re-deposition of nanoparticles without further interpretation and correlative performance evaluation under the operating conditions (*Nat. Commun.* **2018**, 9, 1011; *Nat. Mater.* **2022**, 21, 859–863). Among the seldomly reported works that investigate the behavior and dynamics of the nanoparticles in their operating electrochemical environments, the measurement precision in electrochemical signals is highly limited, making it challenging to establish the connection between the structure change and the electrochemical processes (*Nano Lett.* **2018**, 18, 1280–1289; *ACS Nano.* **2019**, 13, 11372–11381; *Energy Environ. Sci.* **2019**, 12, 2476–2485; *ACS Energy Lett.* **2023**, 8, 1929–1935).

In our in-situ observation process, we obtained a single complete CV with a full range from hydrogen ad/desorption to electric double layer and oxygen ad/desorption within TEM; the results show a typical CV curve with pronounced oxidative and reductive peaks. Meanwhile, changes near the surface of the Pd@Pt electrocatalyst in individual CV cycles were captured, and the unprecedented structure details and change in element distribution reveal that the surface reconstruction was driven by the hydroxide/oxygen adsorbates.

Reviewer #3:

In this study, the authors employed in-situ liquid cell TEM to observe the electrochemical corrosion behaviors of Pd@Pt nanoparticles. The in-situ TEM images provide some information during the corrosion process that are difficult to capture using ex-situ methods, but at a significantly reduced resolution. The proposed mechanism lacks sufficient support from the data and evidence presented. Furthermore, the significance and novelty of this process/mechanism do not appear to be groundbreaking enough to warrant publication in Nature Communications. I suggest considering an alternative journal for publication, subject to the clarification of several key questions.

Response:

Thanks for the reviewer's valuable suggestions. In this work, we use TEM with an electrochemical cell to image the structural changes of the catalyst during its use, which is key to building the correlation between structure and the electrochemical process. Because a thick layer of electrolyte solution is required for such set-up, electron scattering by the liquid layer can limit the spatial resolution. However, the combination of in-situ and ex-situ imaging still helps to resolve the intermediate structure at atomic resolution.

Based on the suggestions, we have made a number of changes, and we provide a point-to-point response, which is shown below. To make the changes easier to identify, the changes to our manuscript were highlighted in yellow.

Comment #1: *"Due to its low resolution of the Supplementary Fig. 4, it will be more convincing if it has similar magnification and resolution as those in Figs. 1d-g, to confirm that the core-shell structure indeed transformed to PdPt alloy nanoparticles."*

Response: Thanks for the reviewer's suggestion. We added the higher magnification EDS mapping to confirm the formation of PdPt alloy nanoparticles after in-situ electrochemical CV (-0.5 V to 0.5 V vs. Pt). The manuscript is revised as below:

Supplementary Fig. 4 HAADF-STEM and EDS characterizations of irregular materials after in-situ electrochemical CV (-0.5 V to 0.5 V vs. Pt) treatment. a, e, Atomic-resolution STEM images of a Pd@Pt octahedron after in-situ electrochemical CV (-0.5 V to 0.5 V vs. Pt) treatment. b-d, f-h, EDS mapping of a Pd@Pt octahedron after in-situ electrochemical CV (-0.5 V to 0.5 V vs. Pt) treatment. The green and red colors correspond to Pd and Pt elements, respectively.

Comment #2: "In the oxidation period, e.g., stage 1, if the corrosion process follows the proposed mechanism, there should be Pt oxides forming at the corner/edge, which can be and need to be confirmed by experimental characterizations, for example, the lattice distance in the images, either the in-situ or ex-situ images, or EDS and EELS, etc. In the manuscript, no oxides are evidenced in the experimental data, making the proposed corrosion process and mechanism less convincing."

Response: Thanks for the comment. Limited by in-situ liquid TEM techniques, it is difficult to realize EELS measurements of nanoparticles in liquid cell. The thickness of liquid layer is thick, at about 500 nm, the liquid causes strong electron scattering, and in EELS, the broad plasmon peak from the liquid obscures signals from the sample, chemical, and valence analysis is very challenging (*ACS Nano* **2021**, 15, 10228–10240; *Adv. Funct. Mater.* **2022**, 32, 2105; *Nat. Nanotechnol.* **2011**, 6, 695). Considering the small amount of Pt oxide formed on the surface, detection of such species is very difficult.

The Pourbaix diagram (Fig R2) of Pt shows that platinum oxides will be generated at high potentials in our in-situ experiments (Voltage up to 1.05V, pH=1). We can determine that peak 2 in Fig. 3 corresponds to the oxidation peak of surface Pd, and the difference in the oxidation potentials between Pd and Pt is exactly about 0.13 V (Lange's Handbook of Chemistry). Therefore, we can determine that peak 3 belongs to the oxidation peak of Pt. Several articles also reported that Pt will be oxidized to form Pt oxides during electrochemical testing (*Acc. Chem. Res.* **2013**, 46, 1848–1857; *ACS Catal.* **2021**, 11, 9904–9915; *ACS Catal.* **2019**, 9, 8622–8645), has been cited in the manuscript.

Fig R2. Pourbaix diagram of Pt in 10% HNO₃ medium.

Comment #3: "In Fig. 2a (605 seconds), the "new island" indicated by red triangles might not originate from the corrosion of the highlighted particle. Instead, it appears to be from a particle moving in from the lower left corner, as observed in the movie. The image below, captured from the movie at 9:50, shows the particle from the lower left corner approaching the central crystal, with the "island" located in between marked by red arrows. Tracking the formation of the "island" throughout the entire movie suggests that it is unlikely to have derived from the central crystal."

Response: Thanks for the reviewer's suggestion. After the oxidative etching, here the dissolved palladium ions in the solution were reduced and deposited on the nanoparticles. The source of the palladium island can be all the nanoparticles nearby, not only from the one in the center.

Comment #4: "Supplementary Fig. 12 shows obvious crystal morphology evolution (indicated by red circles), resembling the change in crystal morphology observed in Figure 2 under electrochemical conditions. This finding implies that the electron beam could play a role in altering the crystal morphology."

Fig 2. a, Time-sequential in-situ TEM images of morphological evolution under the CV from -0.2 V to 0.2 V vs. Pt. b, Time-sequential in-situ TEM images of morphological evolution under the CV from -0.9 V to -0.2 V vs. Pt. The red and green dotted areas are named Island a and Void b, respectively.

Supplementary Fig. 12 Time-sequential in-situ TEM images of Pd@Pt octahedral nanoparticles without electrochemical conditions showed that the electron beam didn't damage the materials. The scale bar was 50 nm. The dose rate we used in all the in-situ experiments was $50 \sim 60 \text{ e}/\text{\AA}^2 \cdot \text{s}$.

Response: Thanks for the comment. Before the in-situ electrochemical experiments, we adjusted the beam dose rate ($50 \sim 60 \text{ e}/\text{\AA}^2 \cdot \text{s}$) to ensure the electron beam would not damage our samples. We observed that most of the particles were very stable under the long-time radiation with this electron dose, and the surfaces remained flat. The changes noticed by the reviewer in a few particles were due to the rotation of the particles in the liquid. Nanoparticle rotation will change their projected 2D images (*Science*, **2020**, 368, 60-67; *Science*, **2015**, 349, 290-295; *Nat. Commun.* **2018**, 9, 421; *Nano Lett.* **2019**, 19, 2871–2878).

Comment #5: "In Fig. 3, it is suggested to use visually different colors to represent Pd, Pt, Pd ion, and Pt ion for a clearer illustration."

Response: Thanks for the reviewer's suggestion. We have modified the atomic color scheme for a clearer illustration. The revised content is highlighted in yellow in the revised manuscript and listed below:

Fig. 3 | CV analysis of the evolution process. *a*, the in-situ CV curve corresponds to the electrochemical process in Fig 2c, and the illustrations of surface corrosion correspond to the redox peaks. *b*, The adjacent CV curves showing the dynamic of the electrochemical corrosion process and the enlarged regions of the corresponding redox peaks. *c*, Thematic illustration of the oxidation potential difference between surface Pd, Pt, and inner Pd atoms.

Fig. 5 | Schematic diagram of the mechanisms of electrochemical corrosion via ex-situ and in-situ paths, respectively. Scale bars: 20 nm.

Comment #6: "Please correct typos throughout the manuscript. For example, in Page 3, "combination with aberration and ..." should be "combination with aberration correction and ..."; In Supplementary Fig. 9, "green circle" in the legend should be "green rectangle"."

Response: Thanks for the reviewer's valuable suggestions on improving the quality of our manuscript. We checked and corrected similar issues throughout the entire manuscript. The revised content is highlighted in yellow in the revised manuscript and listed below:

On Page 3, Para 1, line 14: " In combination with aberration and" has been revised to "In combination with aberration correction and"

On Page 28, Para 2, line 7: " Celebration of RE" has been revised to "Calibration of RE"

In Supplementary Fig. 9: " Fig. 9b, showing the etching region of he inner Pd atoms (green circle)." has been revised to "Fig. 9b, showing the etching region of the inner Pd (green rectangle)."

Reviewer #4:

The work by Wu and co-workers used in situ electrochemical liquid-cell TEM to track the morphological change during oxidative corrosion of Pd@Pt nanoparticles. Overall the experiments well performed with decent quality of the electrochemical data acquired in liquid-cell TEM. The reviewer points out the following critical questions should be addressed before this work can be considered for publication in Nature Commun.

Response:

Thanks for the reviewer's suggestions. We have corrected these mistakes and modified our description. Here, we provide the point-to-point response, shown below. Changes made to our manuscript are highlighted in yellow.

Comment #1: "Interpretation of CV profiles in Fig.3, such as assignment of two oxidative peaks, require reference data of CV profiles of Pd@Pt core-shell NPs in standard 3-electrode electrochemical cell."

Response: We added the data for the calibration of the Pt wire reference electrode using RHE reference electrodes (Supplementary Fig. 13). By comparing the in-situ and ex-situ experiments, we identified that peak 2 in Fig. 3 corresponds to the oxidation peak of surface Pd, and the difference in the redox potentials between Pd and Pt is exactly 0.13 V (Lange's Handbook of Chemistry). Therefore, peak 3 corresponds to the oxidation peak of Pt. The revised content is highlighted in yellow in the revised manuscript and listed below:

On Page 27, Para 2, line 12: "ensure the electron beam would not damage the materials (Supplementary Fig. 12)." has been revised to "ensure the electron beam would not damage the materials (Supplementary Fig. 12). To calibrate the Pt reference electrode, the same normal CV curves of commercial Pt/C catalysts were measured using the reversible hydrogen electrode (RHE) and Pt wire as reference electrodes. (Supplementary Fig. 13)."

Supplementary Fig. 13 has been added:

Supplementary Fig. 13 Calibration of Pt wire reference electrode using RHE.

Comment #2: "The potential of Pt pseudo-RE in 0.1M acid needs calibration against reversible hydrogen electrode (RHE). It is very important to know the potential of Pd/PdOx and Pt/PtOx on the RHE scale and compare to literature reports in electrochemistry/catalysis community."

Response: We added the data for the calibration of the Pt wire reference electrode using RHE reference electrodes (Supplementary Fig. 13). The Pourbaix diagram (Fig R2) of Pt shows that platinum oxides form at high potentials. However, In the in-situ liquid cell, liquid flow and solute diffusion can be affected by the confines of the nanochannel, resulting in overpolarization, potential drift, and electrochemical curve distortion (*ACS Nano* **2015**, 9, 4379–4389; *Sci. Sin. Chim.* **2021**, 11, 1480-1500). The number of internal active particles in the in-situ liquid cell can differ by several orders of magnitude from that of the actual macroscopic electrolytic cell. The interaction between catalysts, electrodes, electrolytes, and the ionic conduction conditions are also different. Therefore, the absolute value of the oxidation potential in the in-situ experiments is different from the values from ex-situ (*Acc. Chem. Res.* **2016**, 49, 2015–2022; *Adv. Funct. Mater.* **2022**, 32, 2105188). Thus, before we performed our experiments, we waited until the electrical signals were stable.

Fig R2. Pourbaix diagram of Pt in 10% HNO₃ medium.

Comment #3: "The reviewer found that analysis of contrast based on TEM images (Fig. 2) is limited to extract reliable information. STEM imaging in liquid is highly recommended to directly show the etching of Pd vs. Pt as those two elements have very different Z contrast."

Response: Thanks for the reviewer's suggestion. We notice that the liquid layer is about 500 nm in thickness (determined by the spacer placed in between the window membranes) in our electrochemical liquid cell TEM, and the bowing of the window due to the pressure difference between the chips gap and TEM column can make the liquid layer thicker. The strong electron scattering can significantly affect the image quality regardless of the imaging mode (*Adv. Funct. Mater.* **2022**, 32, 2105188; *MRS Bull.* **2015**, 40, 46–52; *Nanoscale.* **2020**, 12, 22192-22201). The strength of TEM imaging is its larger depth of focus and higher acquisition rate, which allows us to capture changes in the in-situ

experiments. To analyze the species and their distribution, STEM imaging was performed on the samples left on the E-Chips after we paused and completed the in-situ experiments.

We notice that the acquisition rate and image signal/noise in STEM imaging have improved significantly due to the advancement of new electron detectors. In future experiments, we will try to perform in situ liquid cell imaging using STEM mode.

REVIEWERS' COMMENTS

Reviewer #1 (Remarks to the Author):

The author addressed all my comments. I would recommend its publish, thanks.

Reviewer #2 (Remarks to the Author):

The authors have addressed the comments raised. I recommend publishing.

Reviewer #3 (Remarks to the Author):

The revised manuscript has addressed most questions and concerns raised during the first round of review. I therefore recommend it to be accepted for publication.

Reviewer #4 (Remarks to the Author):

The authors have addressed the reviewer's questions. The reviewer recommends its publication.